# Evaluation of Natural and Vaccine-Induced Anti-SARS-CoV-2 Immunity: A Comparative Study between Different Groups of Volunteers

**DOI:** 10.3390/diseases10020025

**Published:** 2022-04-27

**Authors:** Maria Caterina Schipani, Flaminia Tomassetti, Isabella Polidori, Paola Ricci, Maria Loredana Frassanito, Silva Seraceni, Maria Morello, Eleonora Nicolai, Stefano Aquaro, Sergio Bernardini, Massimo Pieri, Graziella Calugi

**Affiliations:** 1Lifebrain Cosenza—Bilotta, 87100 Cosenza, Italy; mariacaterina.schipani@lifebrain.it (M.C.S.); paola.ricci@lifebrain.it (P.R.); 2Department of Pharmacy, Health and Nutritional Sciences, University of Calabria, 87100 Cosenza, Italy; aquaro@uniroma2.it; 3Department of Experimental Medicine, University of Tor Vergata, 00133 Rome, Italy; flaminia.tomassetti@students.uniroma2.eu (F.T.); morello@uniroma2.it (M.M.); nicolai@med.uniroma2.it (E.N.); bernards@uniroma2.it (S.B.); 4Lifebrain s.r.l., Viale Roma 190/A, 00012, Guidonia Montecelio, 00133 Rome, Italy; isabella.polidori92@gmail.com (I.P.); frassalore@gmail.com (M.L.F.); graziella.calugi@gmail.com (G.C.); 5R.D.I. Lifebrain, Limena, 35010 Padua, Italy; silva.seraceni@lifebrain.it; 6Clinical Biochemistry, Tor Vergata University Hospital, 00133 Rome, Italy

**Keywords:** vaccine, SARS-CoV-2, antibodies, immunology

## Abstract

(1) Background: The production of anti-SARS-CoV-2 antibodies should help minimize the severity of COVID-19 disease. Our focus was to investigate and compare different vaccination schedules, monitoring circulating S-RBD Ab (antibodies anti—Spike protein—Receptor Binding Domain) levels after administering two doses in naïve patients. Likewise, vaccine-stimulated immunity in naïve and previously infected patients was compared. (2) Methods: We included 392 patients. Sera were evaluated by Elecsys anti-SARS-CoV-2 S. Statistical analyses were conducted by MedCalc and JASP. (3) Results: In COVID-19 patients, the median value of Ab levels was 154 BAU/mL, stable up to 9 months after the infection. From the data observed in vaccinated patients, higher median values were recorded in COVID-19/Pfizer BioNTech (18913 BAU/mL) than in other groups (Pfizer BioNTech: 1841; ChadOx1 961; heterologous vaccination: 2687) BAU/mL. (4) Conclusions: In conclusion, a single booster dose given to previously infected patients raised an antibody response much higher than two doses given to naïve individuals and heterologous vaccination generated a robust persistent antibody response at high levels, steady up to three months after administration.

## 1. Introduction

Understanding the immune response to the SARS-CoV-2 is critical to improve diagnostic pathways and vaccine platforms and providing perspective on the future course of the pandemic. Antibodies are the only immune memory component able to provide a “sterilizing” immunity through neutralizing antibodies (nAbs) that can block the virus even before the infection of our cells [1]. Nowadays, different kinds of commercial kits are available to analyze various circulating immunoglobulins produced versus the virus proteins, such as Anti-Nucleocapsid (N) or Anti-Spike-SARS-CoV-2 antibodies (S-RBD Ab). In this work, the focus is on the circulating S-RBD antibodies.

The production of anti-SARS-CoV-2 antibodies should prevent the virus enter to the upper respiratory and oral cavity cells and would minimize the severity of COVID-19 disease to a regular cold or asymptomatic disease [2]. This finding is the main focus of current clinical trials of COVID-19 vaccines. Although the mechanism of protection has still to be clarified [3], a recent study showed a strong correlation between circulating levels of anti-SARS-CoV-2 Spike IgG antibodies and the symptom onset after a doubled-dose vaccination [4].

Since 1 January 2021 in Italy, the first doses of BNT162b2 vaccine (Pfizer/BioNTech, Comirnaty) were delivered [5]. After a short period, the ChAdOx1 nCoV-19 vaccine was distributed to the population. Both have a double-dose administration, the prime and boost approach, but with different modalities and principles. Strong evidence has been reported to achieve a high immunity, which was demonstrated, for example, to be effective in preventing 95% of COVID-19 cases [6].

However, after the European Medicines Agency (EMA) linked the ChAdOx1 nCoV-19 vaccine with rare, yet severe and sometimes fatal, adverse thromboembolic events mainly in younger people [7], Italy and other countries in Europe halted their distribution of this vaccine to either parts or all of their population, recommending a first-dose ChAdOx1 nCoV-19 vaccine followed by the BNT162b2 as second dose, called heterologous vaccine [8]. Current studies suggest that heterologous vaccine built a robust immune response [7,9], which appears to be similar to or even greater than homologous (BNT162b2 or ChAdOx1 nCoV-19) vaccine response. Moreover, others studied the S-RBD Ab level after vaccination following infection, which may provide stronger protection than either natural or vaccine-induced immunity alone [10]. This last one is called hybrid immunity.

Our aim was to investigate different vaccination schedules and compare antibody levels, between BNT162b2, ChAdOx1, and heterologous schedule, monitoring S-RBD Ab levels after administering two doses in naive patients. Likewise, vaccine-stimulated immunity in naive and previously infected patients was compared to natural immunity developed in mild or moderate symptoms patients.

## 2. Materials and Methods

### 2.1. Patients

All the sera were obtained from private clinical laboratory analysis LIFEBRAIN COSENZA Srl, Cosenza (CS), Italy, collected in the period May–September 2021. At the time of laboratory acceptance, short interviews were conducted to enroll patients in the study and to divide them into different groups, as described below. Patients previously infected with COVID-19 joined group 1. Group 2, 3, and 4 patients were selected after they completed a questionnaire, declaring that they had never been infected by SARS-CoV-2 before vaccination. Moreover, all the samples were collected after the second dose administration. In group 5, patients were included who had previously been infected with COVID-19 and with one-single booster dose of Pfizer BioNTech. Table 1 summarizes the principal characteristics of each group, as anamnestic data, division between infection or vaccine, the number of subjects enrolled, the period spend after infection or the vaccination, and the monitored period.

The sera collected were centrifugated for 10 min at 2000× *g* and then processed as fresh samples.

We included 392 patients, 185 males and 207 females, aged 10 to 87 years. All patients with Anti-SARS-CoV-2-S antibody values < 0.82 BAU/mL were excluded from the study. Table 2 details the inclusion and exclusion criteria adopted to enroll in the study volunteer subjects.

The study was approved by the Ethical Committee of University of Rome “Tor Vergata”, (approval number: R.S.44.20). Informed consent was obtained from all the subjects enrolled in the study. The study has been conducted following the Declaration of Helsinki (2013).

### 2.2. Antibody Dosage Anti Spike-SARS-CoV-2

The Elecsys anti-SARS-CoV-2 S (Roche S-RBD tAb) is an electro-chemiluminescence immunoassay (ECLIA) for the in vitro quantitative determination of total antibodies (IgG/IgA/IgM) to the SARS-CoV-2 S-RBD protein in human serum and plasma, performed by fully automated Roche Cobas E602 analyzer (Roche Diagnostics GmbH, Mannheim, Germany).

The assay is a three-step test process that uses a recombinant protein representing the RBD of the S antigen in a double-antigen assay format, which favors the detection of high-affinity antibodies against SARS-CoV-2. Patient samples are incubated with a mix of biotinylated and ruthenylated RBD antigens to form double antigen immune complexes. After the addition of streptavidin-coated microparticles, DAGS complexes bind to the solid phase. The reagent mixture is then transferred to the measuring cell, where the microparticles are magnetically captured. Electrochemiluminescence is induced by applying a voltage and measured with a photomultiplier. The signal yield increases with the antibody titer. The cut-off value in arbitrary units (AU/mL), the conversion factor to obtain BAU/mL, the cut-off value in BAU/mL and the linearity range in AU/mL are respectively: 0.8, 1.0288, 0.823, and 0.40–250, as declared by the manufacturer.

Samples with values over 250 AU/mL (257.2 BAU/mL) were diluted and measured at 1:50.

### 2.3. Statistical Analysis

For normal data distribution, parametric tests were used, such as analysis of variance with Bonferroni post hoc test for more than two variables. For the not-normal data distribution non-parametric test, the Kruskal–Wallis test (variables with more than two categories) was used to verify differences between groups. All statistical analyses will be performed using MedCalc Version 18.2.18 (MedCalc Software Ltd., Ostend, Belgium) and JASP 0.15.0.0 (JASP program, University of Amsterdam, Amsterdam, The Netherlands).

## 3. Results

The patients were divided into five groups, as illustrated in Table 1. The criteria for inclusion and exclusion are summarized in Table 2. Box whisker plots (Figure 1 and Figure 2) were used to graphically represent the distribution of the data. The data were not normally distributed (verified by applying the Shapiro–Wilk test) and were characterized by median and percentiles values.

### 3.1. COVID-19 Infected Patients

Figure 1 shows the persistence of antibody anti-SARS-CoV-2 levels in patients affected by COVID-19 (group 1), who were tested from 1 to 9 months after infection with SARS-CoV-2. The patients analyzed were mostly in an age range of 40–50 years old (age mean 43.36, SD 17.74), with no significant prevalence of morbidity in males, 43% (57 out of 133), compared to females, 57% (76 out of 133). The antibody levels were very heterogeneous (median 154 BAU/mL) with a minimum value at 6 months post-infection of 3.5 BAU/mL and a maximum value of 3773 BAU/mL. The antibodies increase began to be observed since the first month after the disease, with maximum values between the fourth and fifth months. A moderate antibodies decrease was observed in the following months. The data showed that antibody levels persist over time remaining relatively stable up to 9 months after infection (median 107.4 BAU/mL; Interquartile Range (IQR): 81.6–291.1 BAU/mL). All data showed no significance (*p*: ns).

### 3.2. Trend over Time for Each Group

To assess levels of ANTI-SARS-CoV-2 antibodies in vaccinated patients (Figure 2), blood samples were collected from 172 Pfizer BioNTech patients (group 2), 38 ChAdOx1 nCoV-19 patients (group 3), 32 heterologous vaccination patients (group 4), and, lastly, 26 previously infected with a single booster of Pfizer-BioNTech patients (group 5).

As can be seen from Figure 2A, the antibody levels in group 1 declined rapidly in the first month after the vaccination plan and then stabilize over time. The median value of the first month highlighted an antibody peak at 3744 BAU/mL (IQR: 2064–7948 BAU/mL), which significantly (*p* < 0.01) declined by 39% in the second month, 2271 BAU/mL (IQR: 1366–3479 BAU/mL) and dropped by 66% at the third month (*p* < 0.01) 1280 BAU/mL (IQR: 682–1904 BAU/mL).

Despite the Pfizer-BioNTech vaccine, Figure 2B illustrates that ChadOx1 nCoV19 antibody levels after 30 days from second dose administration were significantly lower (*p* < 0.01). The median value obtained, 577 BAU/mL (IQR: 464–1814 BAU/mL), is 6.5 times lower than Pfizer BioNTech median value. However, after 2 months of full schedule administration, the median value raised to 1106 BAU/mL (IQR: 512–1815 BAU/mL), still 2 times lower than Pfizer BioNTech median at 2 months (*p* < 0.01). In the third month, the median value settled at 327 BAU/mL (IQR: 211–1598 BAU/mL), 74% lower than Pfizer BioNTech third month median value.

In Figure 2C, antibody levels of heterologous vaccine patients have been shown; after the first month after booster administration, the median value, 5617 BAU/mL (IQR: 2625–8992 BAU/mL), was 1.5 times higher than group 2 (*p* = ns) and 10 times higher than group 3 (*p* < 0.01). After 2 months, the median value achieved, 2572 BAU/mL (IQR: 1820–4273 BAU/mL), was 2 times higher than group 3 (*p* < 0.01). No significant differences were noted in the second month between group 4 and group 2 median value (*p* = ns). In the third month, a median value of 3715 BAU/mL (IQR: 2202–4941 BAU/mL) was observed; 2.6 times higher than group 2 (*p* < 0.01) and 10 times higher than group 3 (*p* < 0.01).

Regarding the last group, in Figure 2D, the results obtained from patients with previous SARS-CoV-2 infection were analyzed; these patients were given a single dose of Pfizer-BioNTech vaccine after a period ranging from 4 to 10 months after infection. These patients were tested after a COVID-19 negative result and before booster administration (Time 0, T0). At T0, the median value was 142 BAU/mL (IQR: 80–199 BAU/mL), not shown in the figure. After one month of boosts administration, antibody levels median value raised to 22,374 BAU/mL (IQR: 12,596–25,720 BAU/mL). Furthermore, it is possible to observe the antibody levels slightly decline during the second and the third month, maintaining very greater values, 18,203 BAU/mL (IQR: 16,064–22,671 BAU/mL) and 13,727 BAU/mL (IQR: 7097–21,515 BAU/mL), respectively. All group 5 data were significantly different when compared to each month of every group (*p* < 0.001).

### 3.3. Comparison of Antibody Levels over Time between All Groups

Figure 3 shows overall S-RBD antibody levels from vaccinated patients for each group (3A) and divided for the 3 months (3B). In violin bow Figure 3A, it was highlighted that group 5 S-RBD Ab distribution was significantly greater than other groups (*p* < 0.01). Higher median values were recorded in group 5, 18,913 BAU/mL (IQR: 11,447–25,720 BAU/mL) than in other groups. The median value of Pfizer BioNTech was 1841 BAU/mL (IQR: 1158–3523 BAU/mL); the median value of ChadOx1 was 961 BAU/mL (IQR: 347–1758 BAU/mL); and the median value of ChadOx1/Pfizer BioNTech was 2687 BAU/mL (IQR: 2060–5205 BAU/mL). Since group 5 wide distribution, in Figure 3B, the S-RBD Ab levels of each group were compared for every month at the same *y*-axis scale. Evaluating group 5 antibody levels at 1 month, it was observed that its median value, 22,374 BAU/mL (IQR: 12,596–25,720 BAU/mL), was 6.5 times higher than group 2, 3744 BAU/mL (IQR 2064–7948 BAU/mL; *p* < 0.001), 39 times higher than group 3, 577 BAU/mL ((IQR: 464–1290; *p* < 0.001), and 4 times higher than group 4, 5617 BAU/mL (IQR: 2625–8991 BAU/mL; *p* < 0.001). After two months, group 5 median value was 18,203 BAU/mL (IQR: 16,064–22,671 BAU/mL) higher than group 2, 2271 BAU/mL (IQR: 1366–3479 BAU/mL; *p* < 0.001); 16.5 higher than group 3, 1106 BAU/mL (IQR: 512–1815 BAU/mL; *p* < 0.001); and 7 times higher than group 4, 2572 BAU/mL (IQR: 1820–4273 BAU/mL). Three months later vaccine administration was observed in group 5 a response, median value 13,727 BAU/mL (IQR: 7097–21,515 BAU/mL), 11 times higher than group 2, 1288 BAU/mL (IQR: 682–1904 BAU/mL; *p* < 0.01); 42 times higher than group 3, 327 BAU/mL median value (IQR: 211–1598 BAU/mL); and 4 times higher than group 4, 3715 BAU/mL median value (IQR: 2202–4941 BAU/mL).

## 4. Discussion

The immunity duration in people who have tested positive for COVID-19 nowadays is one of the most discussed topics in the scientific world and, to date, there is still no definitive and unambiguous answer in this regard. What we know is that following primary infection, over 90% of patients develop a positivity for antibodies against SARS-CoV-2, which, although decreasing over time, remain detectable for several months [2,11].

In the present work, the circulating S-RBD Ab persistence was analyzed in 122 patients who contracted the SARS-CoV-2 virus, up to 9 months after infection. A retrospective analysis of the data revealed a quite heterogeneous antibody response, which could depend on the severity of the disease (asymptomatic, mild, moderate, or severe) and inter-individual variability [2,9]. However, the antibody levels persist over time, relatively stable up to 9 months after infection, at a medium-low level (median 107.4 (81.6–291.1) BAU/mL) (Figure 1). Furthermore, in some patients, the circulating S-RBD Ab levels are even increased over time, which contrasts with other recent publications that have found a slight decrease in the antibody levels [2,10]. This could have several explanations. Firstly, the test used evaluates total Ig rather than IgG, which could mean that test evaluates all the circulating immunoglobulin classes (IgA, IgG, and IgM). Second, the S-RBD Ab test inherently favors the detection of high-affinity antibodies [11]. Therefore, the increase in signal in the RBD test may reflect the maturation of antibody affinity over time, while the total amount of S-RBD Ab effectively remains stable or even slightly decreases [12]. The results obtained in this study are consistent with the data obtained in recent papers [2,6,9,13], where the persistence of circulating antibodies against the Spike protein and/or the domain RBD was observed in convalescent patients for 5, 6, 8, and 11 months, respectively, after the onset of symptoms. 

For vaccinated patients, the data obtained in this work showed antibody levels decline after the immunological peak (Pfizer-BioNTech after 21 days, ChAdOx1 nCoV19 after two months), while from the fourth month onwards, the titer stabilizes over time. However, despite the first vaccine, the ChAdOx1 nCoV19 antibody response seemed more contained. After one month, the level of circulating antibodies S-RBD Ab is significantly lower than Pfizer-BioNTech. Otherwise, heterologous vaccination elicited a robust antibody response. Comparing the results obtained with homologous vaccination patients, a statistically significant difference was observed in circulating S-RBD Ab levels, even after 30 days. Still, after three months from the administration of the booster, the heterologous antibody levels continued to be significantly higher than Pfizer-BioNTech and/or ChAdOx1 nCoV19. The data obtained in the present work, therefore, provide evidence of strong immunogenicity of the heterologous vaccination capable of generating a robust persistent antibody response at high levels steady up to three months after administration. Several hypotheses explain these disparities between the groups, including differences in the age-populations tested, the different lifestyles, and biological variability of immune response after the SARS-CoV-2 vaccine. Moreover, the specific and technical properties of vaccine composition could lead to the results obtained [14].

From the data observed in group 5, the administration of one-single dose of booster developed a stronger antibody response, much more if compared to other groups. Several studies were in agreement with these findings [13,15], highlighting that a single booster dose given to previously infected patients raised an antibody response much higher than two doses given to naïve individuals. Furthermore, monitoring the antibodies levels after vaccine administration or after the booster dose is the key to better understanding the overtime antibodies stability and to determine if their protective action is level dependent. It should be also considered whether periodic booster doses could or could not overstimulate the immune systems [16] and these studies should investigate the cell immune memory response, too [17].

The assertion on the most effective vaccination strategy as the heterologous one is referred to naïve patients [2,9]. When the infection occurs the effect on immune system stimulation is higher than vaccine alone. Several works have already demonstrated how administration of a single dose on previously infected patients results in higher Abs production respect to double-dose vaccinated patients [14,15].

The limitation of this study is the imbalance in the different groups between the number of enrolled volunteers, based on the Italian vaccine campaign. Nevertheless, despite the limitations, the data obtained are consistent with the literature [18].

Another limit was the short period evaluated for vaccinated people, regardless some subjects, as they were volunteers, did not want to continue antibody monitoring.

## 5. Conclusions

The data obtained highlighted the importance of establishing the right schedule of vaccination. It has been seen that ChadOx1 nCoV19 patients did not develop a proper anti-SARS-CoV-2 immunity, as well as Pfizer BioNTech patients, whose antibody levels have risen as fast as they have decayed. For this, a third dose seems to be increasingly needed to stimulate immunity and to achieve strong (and protective) antibody levels.

On the other hand, data obtained in previously infected patients suggest just a single dose of vaccine may be sufficient. However, further investigations on reactogenicity and antibody kinetics are needed.

## Figures and Tables

**Figure 1 diseases-10-00025-f001:**
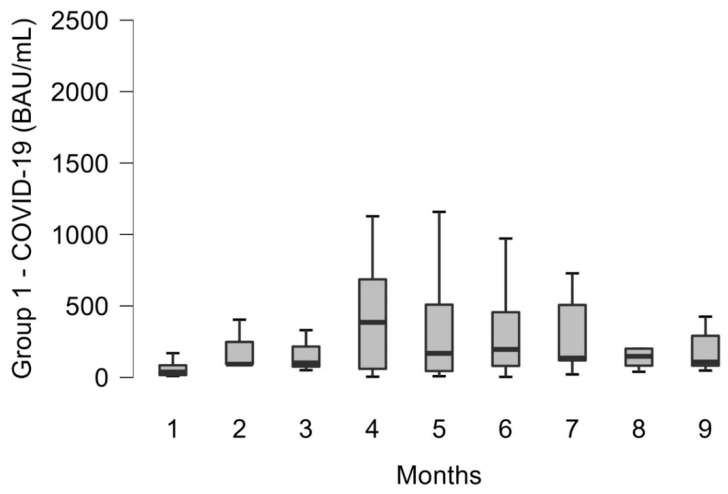
S-RBD antibodies trend of COVID-19 case positive patients (group 1) from 1 up to 9 months after the infection. All the data were not significant between each other (*p*: ns); from the fourth month, an increase is observed which reaches a maximum value after six months (3773 BAU/mL), and a slight decrease in the following months.

**Figure 2 diseases-10-00025-f002:**
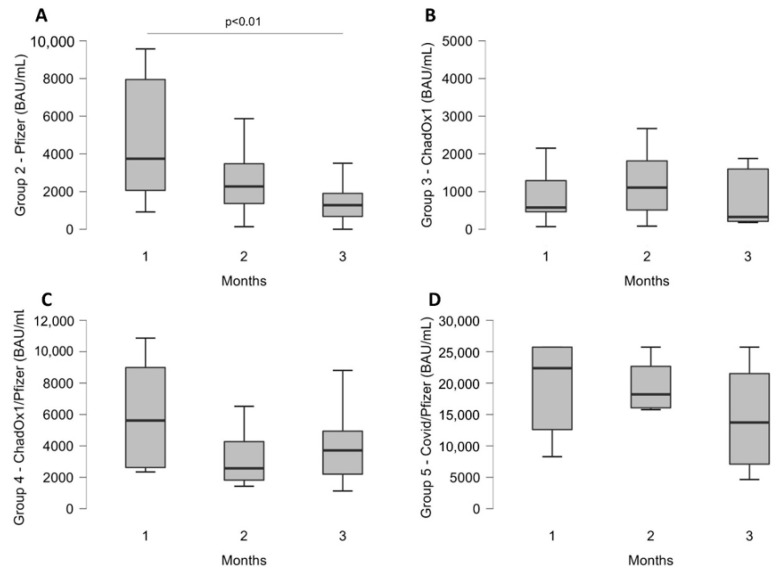
S-RBD antibody levels from vaccinated people were divided as follows: (**A**) Pfizer BioNTech (group 2), the data were significant (*p* > 0.01); (**B**) ChadOx1 (group 3), the data were not significant (*p*: ns); (**C**) heterologous Chadox1/Pifezer BioNTech (group 4), the data were not significant (*p*: ns); and (**D**) previously infected COVID-19/Pfizer BioNTech booster dose (group 5), the data were not significant (*p*: ns).

**Figure 3 diseases-10-00025-f003:**
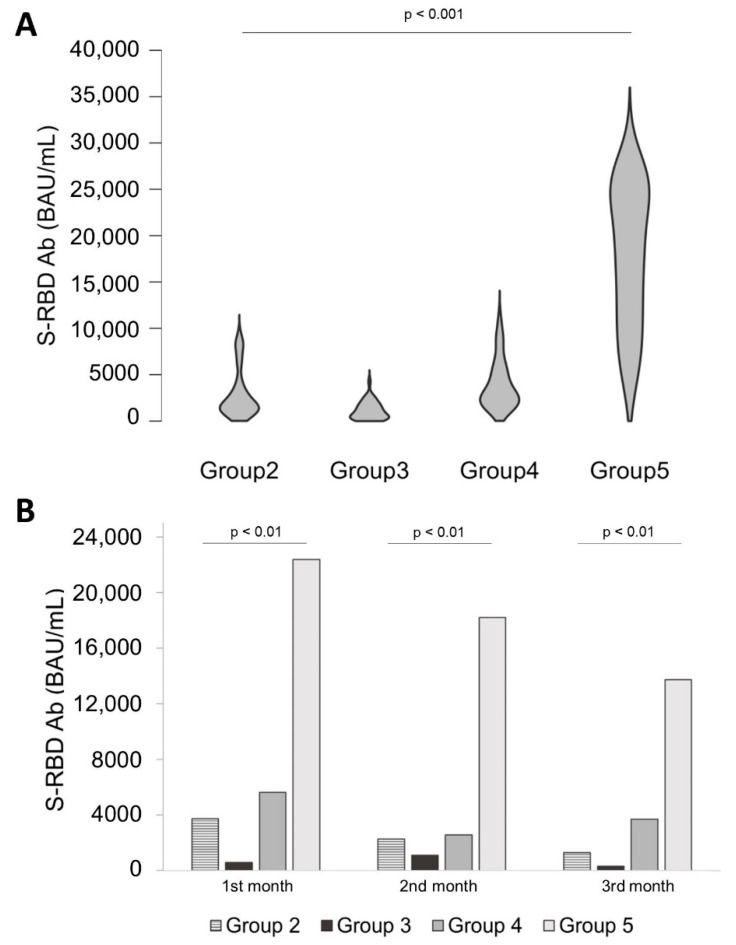
An overall view of all vaccinated groups. (**A**) shows a 3-months S-RBD antibodies distribution for all groups; it is easy to assess intergroup significance (*p* < 0.01). (**B**) is a general monthly histogram-analysis in the same *y*-axis scale of antibodies levels differences between each group. For every month, the vaccinated groups were significantly different (*p* < 0.001).

**Table 1 diseases-10-00025-t001:** Criteria of division and anamnestic information for volunteering subjects, sorted after they completed a questionnaire. [No: number; yrs: years; m: males; f: female].

	Infection/ Vaccine	Age	Gender	No	Schedule	Period	Antibody Monitoring
Group 1	COVID-19 infection	43 ± 18 yrs	m 57 (46 ± 17 yrs) f 76 (41 ± 18 yrs)	133	Previous SARS-CoV-2 infection	April 2020 and September 2021	From 1st month since negative swab to 9 months after
Group 2	Pfizer BioNTech	49 ± 18 yrs	m 85 (53 ± 17 yrs) f 87 (46 ± 17 yrs)	172	Double dose of Pfizer-BioNTech Cominarty (BNT162b2) vaccine	2nd dose: 21 days later	Since administration of 2nd dose
Group 3	ChadOx1 nCoV19	59 ± 13 yrs	m 17 (53 ± 14 yrs) f 21 (63 ± 10 yrs)	38	Double dose of Vaxzevria (ChAdOx1-S) AstraZeneca) vaccine	2nd dose: 4–12 weeks after	Since administration of 2nd dose
Group 4	ChadOx1 nCoV19 + Pfizer BioNTech/heterologous	43 ± 14 yrs	m 16 (43 ± 16 yrs) f 15 (42 ± 10 yrs)	32	First dose (priming) of the vaccine Vaxzevria (ChAdOx1-S) AstraZeneca and a second dose (booster) of the vaccine Cominarty (BNT162b2) Pfizer-BioNTech	Booster dose: 8–12 weeks from the priming (since the ministerial circular of 14 June 2021)	Since administration of booster dose
Group 5	COVID-19 infection + booster dose	44 ± 17 yrs	m 10 (52 ± 15 yrs) f 18 (40 ± 17) yrs	28	Previous SARS-CoV-2 infection and a single dose of anti-SARS-CoV-2/COVID19 vaccine (in this study with Cominarty (BNT162b2) Pfizer-BioNTech preparation)	Vaccine between 4–8 months since infection	Since administration of vaccine

**Table 2 diseases-10-00025-t002:** Inclusion and exclusion criteria to select and divide the volunteer subject.

Inclusion Criteria	Exclusion Criteria
Over 10 years	Under 10 years
Previous infection of COVID-19 in the last 9 months	Under immunosuppressor therapy (in the last 5 months)
Full schedule vaccination (and type of vaccine)	Under immunomodulator therapy (in the last 5 months)
Less than 30 days from the second dose or the booster dose	Chronic disease
Negative SARS-COV-2 antigen screening test	Immunodeficiency
Ig anti-RBD > 0.823 BAU/ML	Ig anti-RBD < 0.823 BAU/mL

## Data Availability

The data presented in this study are available on request from the corresponding author.

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
