# Peer review of "Evaluation of Natural and Vaccine-Induced Anti-SARS-CoV-2 Immunity: A Comparative Study between Different Groups of Volunteers"

_diseases, 2022, doi:10.3390/diseases10020025_

Round 1

Reviewer 1 Report

This work is to describe "Evaluation Of Natural And Vaccine-Induced Anti-SARS-CoV-2 Immunity: A Comparative Study Between Different Groups Of Volunteers". It is very interesting and important topic. However, there are some points should be revise and check.

Major points:

  1. Please add a figure to describe the inclusion/exclusion criteria in the study setting. Also, please add the related information for the patient groups. 
  2. How to confirm the included patients (group 2, 3, 4) infected/uninfected by SARS-COV-2 before vaccine?
  3. Please add a table to describe the demographic characteristics of included patients in each groups (gender, age, immune status et al.,).
  4. Please describe the limitation of this study.

Minor points:

  1. There are some typo errors. The manuscript was suggested to carefully check and revise. Also, the manuscript was suggested to edit by English-speaking native. 
  2. In the table 1, what is the meaning of "After XX months since negative swab"?

Author Response

Comments from Reviewer1:

This work is to describe "Evaluation Of Natural And Vaccine-Induced Anti-SARS-CoV-2 Immunity: A Comparative Study Between Different Groups Of Volunteers". It is very interesting and important topic. However, there are some points should be revise and check.

Major revision:

Comment 1: Please add a figure to describe the inclusion/exclusion criteria in the study setting. Also, please add the related information for the patient groups.

Response: Thank to the reviewer for pointing this out. We agree with this comment; therefore, we have decided to include this information in Table 1 and Table 2, described in the Results.

Comment 2: How to confirm the included patients (group 2, 3, 4) infected/uninfected by SARS-COV-2 before vaccine?

Response: We thank the reviewer, and our group selection criteria are based on the self-declaration written in the informed consent. The conditions self-declared by the subject were expressed in the informed consent, summarized in Table 2.

Comment 3: Please add a table to describe the demographic characteristics of included patients in each groups (gender, age, immune status et al.,).

Response: Thanks to the reviewer for his/her comment, consequently we detailed the demographic characteristics of each group in the Table 1.

Comment 4: Please describe the limitation of this study.

Response: Thank the reviewer for pointing this out. We agree with this comment; therefore, we have written the limitations in lines 296 to 300 in the Discussion.

Minor points:

Comment 1: There are some typo errors. The manuscript was suggested to carefully check and revise. Also, the manuscript was suggested to edit by English-speaking native.

Response: We thank the reviewer to emerge the typo errors. We corrected them.

Comment 2: In the table 1, what is the meaning of "After XX months since negative swab"?

Response: We thank the reviewer, and we are very sorry for the mistake. It was rectified.

Reviewer 2 Report

Dear authors

The manuscript is very interesting but need some improvement regarding following points

Material methods

The table 1 and details of five groups should be in materials and methods

Discussion section needs more improvement and discussion the results with similar previous studies

Author Response

Comments from Reviewer2:

The manuscript is very interesting but need some improvement regarding following points

Comment 1: The table 1 and details of five groups should be in materials and methods

Response: We thank the reviewer, and we added the description of Table 1 and of anamnestic details in lines 95-98 in the Materials and methods.

Comment 2: Discussion section needs more improvement and discussion the results with similar previous studies.

Response: Thank the reviewer for pointing this out. Therefore, we have improved the discussion with new references in lines 277 to 280.

Reviewer 3 Report

The study by Schipani and colleagues compared the effectiveness of various vaccination protocols against SARS-CoV-2 infection in healthy and COVID-19 patients. Based on the level of circulating antibody against the Spike protein, the authors showed that the most effective vaccination strategy in healthy patients is a heterologous strategy, based on the administration of AstraZeneca vaccine and then Pfizer-BioN-Tech vaccine. However, a single administration of Pfizer-BioN-Tech vaccine to previously infected individuals resulted in the highest antibody production. In general, the manuscript is interesting and well written, however the authors should address the following issues before the final approval of this manuscript for publication.

Major issues:

  1. Vaccine evaluation time 3 months after vaccination is relatively short.It would be useful to have Ab levels assessed 6 or 9 months after vaccination as well.
  2. It would also be beneficial to assess the level of IgM and IgG against Spike protein.
  3. Figures 1 and 2: what do the dots above the columns represent?The dots bring chaos to the charts.
  4. The results in Figure 3B are a repetition of Figure 2. Both figures can be compiled into one. Then it will also be easier to show statistically significant differences on the chart. Moreover, description of the results in section 3.3. is a repetition of section 3.2 in the “Results”.

Minor issues:

  1. Table 1: Group 4: in the second column (infection/vaccine), the AstraZenca vaccine should be the first and the Pfizer-BioN-Tech vaccine the second to maintain an appropriate order of vaccine application.
  2. Some parts of the manuscript require minor linguistic correction.

Author Response

Comments from Reviewer3:

The study by Schipani and colleagues compared the effectiveness of various vaccination protocols against SARS-CoV-2 infection in healthy and COVID-19 patients. Based on the level of circulating antibody against the Spike protein, the authors showed that the most effective vaccination strategy in healthy patients is a heterologous strategy, based on the administration of AstraZeneca vaccine and then Pfizer-BioN-Tech vaccine. However, a single administration of Pfizer-BioN-Tech vaccine to previously infected individuals resulted in the highest antibody production. In general, the manuscript is interesting and well written, however, the authors should address the following issues before the final approval of this manuscript for publication.

Major issue:

Comment 1: Vaccine evaluation time 3 months after vaccination is relatively short. It would be useful to have Ab levels assessed 6 or 9 months after vaccination as well.

Response: Thank the reviewer for pointing this out and we included the limitations of the study in lines 299-300 in the Discussion. However, the 3 months period choice was dictated by the impossibility of some voluntaries to continue the antibody monitoring. So, the time window reported was the only statistically comparable.

Comment 2: It would also be beneficial to assess the level of IgM and IgG against Spike protein.

Response: We agree with this comment because it would have increased the value of this research; nevertheless, the kit available for our study in out laboratory is the one used (determination of anti-S-RBD SARS-CoV-2 total antibodies: IgG/IgA/IgM).

Comment 3: Figures 1 and 2: what do the dots above the columns represent? The dots bring chaos to the charts.

Response: Thanks to the reviewer. The dots shown in the figures above the box plots are the outliers, due to the huge dispersion of the data. According to reviewer we deleted them.

Comment 4: The results in Figure 3B are a repetition of Figure 2. Both figures can be compiled into one. Then it will also be easier to show statistically significant differences on the chart. Moreover, the description of the results in section 3.3. is a repetition of section 3.2 in the “Results”.

Response: We thank the reviewer for this comment.

Nonetheless, the two figures are different and highlight distinct aspects of antibody monitoring. Figure 2 shows the trend over time for the same group (A group 2, B group 3, C group 4, D group 5), to highlight differences statistically significant in the same group. While figure 3B shows differences at the same time versus all groups, to highlight differences statistically significant at the same time.

We modified the subtitle of the related paragraph in the Result section.

Minor issue:

Comment 1: Table 1: Group 4: in the second column (infection/vaccine), the AstraZenca vaccine should be the first and the Pfizer-BioN-Tech vaccine the second to maintain an appropriate order of vaccine application.

Response: We thank the reviewer and we corrected it in the Table.

Comment 2: Some parts of the manuscript require minor linguistic correction.

Response: We thank the reviewer, and we increase the English-form.

Reviewer 4 Report

In this study, the authors measured the antibody levels in the individuals that were either infected with SARS-CoV-2 or vaccinated or vaccinated after the SARS-CoV-2 pre-infection. The results are clear and support their conclusion that the pre-infected individuals display maximum protection after the vaccination compared to the vaccination alone. The study is well designed and is very timely. Only minor revisions would be recommended,

  1. It would be quite revealing visually if the Y-axis is set to the same scale for all the graphs.
  2. Can the authors speculate any mechanism for the results and discuss them?
  3. Did the authors collect any information on whether the vaccinated individuals were later infected with SARS-CoV-2? Such information may be quite revealing.

Author Response

Comments from Reviewer4:

In this study, the authors measured the antibody levels in the individuals that were either infected with SARS-CoV-2 or vaccinated or vaccinated after the SARS-CoV-2 pre-infection. The results are clear and support their conclusion that the pre-infected individuals display maximum protection after the vaccination compared to the vaccination alone. The study is well designed and is very timely. Only minor revisions would be recommended.

Comment 1: It would be quite revealing visually if the Y-axis is set to the same scale for all the graphs.

Response: Thank the reviewer for pointing this out, and we tried to adapt the same Y-axis for all the graphs, however, we have chosen to leave the Y-axis as it is because the different scales help to visualize better the antibody trend in the period analyzed for each group. Anyhow, the graph in Figure 3 shows all the data in the same scale Y-axes.

Comment 2: Can the authors speculate any mechanism for the results and discuss them?

Response: We thank the reviewer and, therefore, we have improved the Discussion with founding of previous studies.

Comment 2: Did the authors collect any information on whether the vaccinated individuals were later infected with SARS-CoV-2? Such information may be quite revealing.

Response: Thanks to the reviewer for his/her comment, as matter of fact, we monitored, during all the study period (3 months), the enrolled subjects through rapid antigen test. Anyone did not contract the SARS-CoV-2 infection.

Round 2

Reviewer 1 Report

I appreciate the efforts that the authors have made in response to my questions and concerns.

Author Response

We thank the reviewer for his/her feedback.

Reviewer 3 Report

The authors addressed all comments.

Author Response

We thank the reviewer for his/her response.